# Angiotensinergic Neurotransmissions in the Medial Amygdala Nucleus Modulate Behavioral Changes in the Forced Swimming Test Evoked by Acute Restraint Stress in Rats

**DOI:** 10.3390/cells10051217

**Published:** 2021-05-17

**Authors:** Camila Marchi-Coelho, Willian Costa-Ferreira, Lilian L. Reis-Silva, Carlos C. Crestani

**Affiliations:** 1School of Pharmaceutical Sciences, São Paulo State University (UNESP), Araraquara, SP 14800-903, Brazil; kkcoelho98@gmail.com (C.M.-C.); willian.costa.ferreira@hotmail.com (W.C.-F.); im.lizreis@gmail.com (L.L.R.-S.); 2Joint UFSCar-UNESP Graduate Program in Physiological Sciences, São Carlos, SP 13565-905, Brazil

**Keywords:** stress, amygdala, angiotensin, depression, rodents

## Abstract

We investigated the role of angiotensin II type 1 (AT_1_ receptor) and type 2 (AT_2_ receptor) and MAS receptors present in the medial amygdaloid nucleus (MeA) in behavioral changes in the forced swimming test (FST) evoked by acute restraint stress in male rats. For this, rats received bilateral microinjection of either the selective AT_1_ receptor antagonist losartan, the selective AT_2_ receptor antagonist PD123319, the selective MAS receptor antagonist A-779, or vehicle 10 min before a 60 min restraint session. Then, behavior in the FST was evaluated immediately after the restraint (15 min session) and 24 h later (5 min session). The behavior in the FST of a non-stressed group was also evaluated. We observed that acute restraint stress decreased immobility during both sessions of the FST in animals treated with vehicle in the MeA. The decreased immobility during the first session was inhibited by intra-MeA administration of PD123319, whereas the effect during the second session was not identified in animals treated with A-779 into the MeA. Microinjection of PD123319 into the MeA also affected the pattern of active behaviors (i.e., swimming and climbing) during the second session of the FST. Taken together, these results indicate an involvement of angiotensinergic neurotransmissions within the MeA in behavioral changes in the FST evoked by stress.

## 1. Introduction

Available pharmacological treatments for depression are only partially effective as the remission rate is only 30% for patients treated with the traditional therapy (monoamine pathways alterations), so that new pharmacological targets for depression treatment have been explored [1,2]. Other relevant aspects are evidence from clinical and preclinical studies correlating stress with the pathogenesis of several psychic disorders, including depression [3]. Indeed, studies in rodents revealed that exposition to acute stressors led to alterations in depression-like behavioral in several tasks, including the forced swimming test (FST) [4,5,6,7]. Despite these pieces of evidence, the neurobiological mechanisms involved in behavioral changes evoked by stressful stimuli are not completely understood.

The renin-angiotensin system (RAS) was first described as an endocrine system, but it has been described that biologically active angiotensins might be synthetized locally in several organs and tissues, including in the brain [8]. The RAS has been subdivided mainly into two axes: (i) the angiotensin-converting enzyme (ACE)/angiotensin II/angiotensin II type 1 (AT_1_) receptor axis and (ii) its counterregulatory axis the angiotensin-converting enzyme 2 (ACE2)/Angiotensin-(1–7)/MAS receptor axis [9,10,11]. The first step for angiotensin II synthesis is the hydrolysis of angiotensinogen to angiotensin I by renin [9,12]. Then, angiotensin I is cleaved to angiotensin II by ACE [9,12]. ACE-independent pathways for the formation of angiotensin II from angiotensin I and angiotensinogen have also been described [9,12]. The angiotensin II is biologically active, and its main effects, including pro-stress action, are mediated by activation AT_1_ receptor [10,11,12,13,14,15,16,17,18]. However, angiotensin II also activates the angiotensin II type 2 (AT_2_) receptor [10,11,12]. Angiotensin-(1–7) is mainly synthetized from the hydrolysis of the angiotensin II by the ACE2, but cleavage of angiotensin I and angiotensin II by other peptidases might also contribute [9,12]. The effects of angiotensin-(1–7) are mainly mediated by activation of the MAS receptor [9,10,11].

Evidence points to the involvement of the RAS in the behavioral responses evoked by aversive threats [13,14,15]. Besides, the RAS has been proposed as a potential new target for depression treatment [1,11,15]. Accordingly, activation of the AT_1_ receptor in the central nervous system has been demonstrated as a prominent mechanism involved in behavioral and physiological responses evoked by aversive situations [10,16,17,18]. Thus, observational studies, case reports and interventional studies revealed antidepressant effect in patients treated with AT_1_ receptor antagonists [1,13,14]. Studies in rodents also revealed that exposition to stress changes the angiotensin II synthesis and the AT_1_ receptor expression in the central nervous system [13,19], and systemic pharmacological treatment with AT_1_ receptor antagonists inhibits depressive-like behaviors evoked by chronic stressors [20,21]. Effect of AT_1_ receptor blockade in depressive-like behaviors in non-stressed animals was also demonstrated [22,23,24]. Conversely, ACE-2/Angiotensin-(1–7)/MAS receptor pathway seem to be a contraregulatory RAS mechanism to pro-stress effects of AT_1_ receptor [10,11,15]. Indeed, previous studies revealed that i.c.v. administration of Angiotensin-(1–7) evoked antidepressant-like effects in non-stressed animals [25,26]. The role of the AT_2_ receptor in physiological and behavioral responses to stress is less understood [27]. Despite these pieces of evidence, the specific brain sites whereby angiotensinergic mechanisms operate to control behavioral responses are poorly understood.

Limbic structures play a prominent role in the control of behavioral responses in the central nervous system [28,29]. In this sense, the medial amygdaloid nucleus (MeA) is part of the limbic system and has been reported as the main amygdaloid subnucleus activated by aversive stimuli [30]. The role of MeA in depressive-like behaviors is supported by evidence that intra-MeA microinjection of tricyclic antidepressants, V_1b_ receptor antagonist, the monoamine oxidase inhibitor antidepressant drug minaprine or estradiol decreased immobility in the FST [31,32,33]. Besides, lesion of catecholaminergic terminals within the MeA inhibited the antidepressant-like effect evoked by systemic injection of tricyclic antidepressants and electroconvulsive shock [34,35]. However, the local neurochemical mechanisms involved in behavioral control by MeA is poorly understood. Besides, a role of the MeA in behavioral changes evoked by stressful experience remain to be addressed.

Angiotensinergic terminals, angiotensinogen, and all angiotensin receptors were identified within the MeA [36,37,38,39]. However, the involvement of MeA angiotensinergic neurotransmissions in behavioral responses evoked by stressful events was never investigated. Thus, in the present study we evaluated the role of AT_1_, AT_2_ and MAS receptors present in the MeA in behavioral changes in the FST evoked by an acute session of restraint stress in rats.

## 2. Materials and Methods

### 2.1. Animals

Sixty-three male Wistar rats (60-days-old, weighing 240–260 g) used in this study were obtained from São Paulo State University (UNESP) Animal Breeding Facility (Botucatu, São Paulo, Brazil). They were transferred to the Animal Facility of the School of Pharmaceutical Sciences/Laboratory of Pharmacology (Araraquara, São Paulo, Brazil) at least seven days before the beginning of the experimental procedures. During the entire experimental period animals were housed in collective plastic cages (four rats/ cage) in a temperature-controlled room at 24 °C with light-dark cycle 12:12 h (lights on between 7:00 a.m. and 7:00 p.m.) having free access to water and food. Complying the Brazilian and international guidelines for animal use and welfare, this study was approved by the Local Ethical Committee for Animal Use of the School of Pharmaceutical Sciences/UNESP (approval # 12/2018).

### 2.2. Surgery Procedure

Animals were subjected to inhalation anesthesia with isoflurane (2%) using a low-flow anesthesia system (Bonther, Ribeirão Preto, São Paulo, Brazil). The scalp was anesthetized with 2% lidocaine, and the skull was surgically exposed. Stainless-steel guide cannulas (26 G, 15 mm long) directed to the MeA were bilaterally implanted using a stereotaxic apparatus (Stoelting, Wood Dale, Illinois, USA). The stereotaxic coordinates for cannula implantation were: +5.6 mm from the interaural, +3.4 mm lateral from the medial suture, −8.2 mm ventral from the skull. All parameters were determined from the Paxinos and Watson [40]. The incisor bar position was set at −3.2 mm. Dental cement was applied to fix cannulas to the skull. After surgery, the animals were treated with a poly-antibiotic formulation (streptomycin and penicillin) to prevent infection (560 mg/mL/kg, i.m.) and a non-steroidal anti-inflammatory drug for post-operation analgesia (flunixin meglumine, 0.5 mg/mL/kg, s.c.). Then, the animals were kept in their collective cages for five days before the trials.

### 2.3. Drug Microinjection into the Brain

Microinjection needles (33 G, Small Parts, Miami Lakes, Florida, USA) used to perform intra-brain microinjections of the drugs were one mm longer than the guide cannulas fixed into the brain. The injection needles were connected to a 2 μL syringe (7002KH, Hamilton, Reno, Nevada, USA) by a polyethylene tube (PE-10). All drugs were injected in a final volume of 100 nL [41,42,43]. To avoid reflux, the needle was left in the guide cannula for 40 s after the microinjection before being removed.

### 2.4. Restraint Stress

For the restraint stress, animals were restrained into cylindric plastic tubes (diameter: 6.5 cm and length: 15 cm) for 60 min. The tubes had ½ inch holes that filled 20% of the tube for ventilation. Each animal was restrained only once to avoid habituation.

### 2.5. Forced Swimming Test (FST)

The test consisted of exposing the animals to a 40 cm high cylinder filled with 30 cm of water (25 ± 1 °C). To assess behavioral despair (depressive-like behavior), the test had two sessions (denominated pre-test and test sessions) [44,45], so as described by Porsolt et al. [46,47]. In the pre-test session, animals were individually placed into the water-filled cylinder for 15 min. Afterwards, they were gently wiped with a towel and returned to their home cages. Twenty-four hours later, the animals went to the test session, where they were re-exposed to the water-filled cylinder for 5 min. The behavior was recorded during the pre-test and test sessions using a camera coupled to a microcomputer (Microsoft LifeCam cinema HD).

Antidepressant drugs cause alterations in behavioral responses during the FST [46,47,48], such as decreased immobility (lack of movement, except necessary movements to keep the animal’s head above the water surface) and increase in latency to first bout of immobility and climbing (vertical movements of the forepaws towards the chamber) and swimming (horizontal movements towards the chamber) behaviors [44,49,50]. Therefore, all these behaviors were manually analyzed in a blind manner with the support of the software X-Plot-Rat (it can be freely downloaded at http://scotty.ffclrp.usp.br/X-Plo-Rat.html accessed on 18 October 2018) [42].

### 2.6. Drugs and Solutions

Losartan potassium (selective AT_1_ receptor antagonist) (Tocris, Westwoods Business Park, Ellisville, Missouri, USA), A-779 (selective MAS receptor antagonist) (Tocris), PD123319 ditrifluoroacetate (selective AT_2_ receptor antagonist) (Tocris) and urethane (Sigma-Aldrich, St Louis, Missouri, USA) were dissolved in saline (NaCl 0.9%). The poly-antibiotic formulation (Pentabiotico; Fontoura-Wyeth, Campinas, São Paulo, Brazil), isoflurane (Isoforine; Cristália, Itapira, São Paulo, Brazil), lidocaine (Harvey Química Farmacêutica Ind. e Comércio Ltd.a., Catanduva, São Paulo, Brazil), and flunixin meglumine (Banamine, Schering Plough, Cotia, São Paulo, Brazil) were used as provided.

### 2.7. Experimental Design

Five days after the stereotaxic surgery, the animals were brought to the experimental room in their own home-cages. All rats were allowed at least 60 min to adapt to the sound and illumination conditions of the room (25 °C and acoustically isolated) before starting the experiment. Afterwards, animals randomly received bilateral microinjections into the MeA of the selective AT_1_ receptor antagonist losartan (1 nmol/100 nL, *n* = 9), the selective MAS receptor antagonist A-779 (0.1 nmol/100 nL, *n* = 10), the selective AT_2_ receptor antagonist PD12319 (0.5 nmol/100 nL, *n* = 8) or vehicle (NaCl 0.9%, 100 nL, *n* = 21) [42,51,52,53]. Ten minutes after the drug microinjection into the MeA, the animals underwent the 60 min session of restraint stress. Immediately after the restraint, the rats were placed individually in the water-filled cylinder for a period of 15 min (pre-test session of the FST). Twenty-four hours later, the rats were submitted to the 5 min session of the FST (test session). The behavior in the FST of a control group (non-stressed, *n* = 15) that was not subjected to the restraint stress was recorded to determinate the behavioral changes evoked by acute restraint stress. Immobility, swimming and climbing time; as well as latency to the first bout of immobility were evaluated during both pre-test and test sessions of the FST.

### 2.8. Histological Determination of the Microinjection Sites

At the end of each experiment, the animals were anaesthetized with urethane (1.2 g/kg, i.p.) and 100 nL of Evan’s blue dye (1%) was bilaterally microinjected into the brain as a marker for the injection site. Afterwards, the brain was removed and postfixed in 10% formalin for at least 48 h at 4 °C. Then, serial 40 µm-thick sections of the MeA region were cut using a cryostat (CM1900, Leica, Wetzlar, Germany). The microinjection sites were identified according to Paxinos and Watson [40] in a light microscope.

### 2.9. Data Analysis

Data were expressed as mean ± standard error mean (SEM). Data were analyzed using the Software GraphPad Prism 7 (GraphPad Software Inc., La Jolla, California, USA). Two-way ANOVA, with treatment as main factor and time as repeated measurement, was used to analyze the behaviors during the pre-test session (first session) of the FST. Behaviors during the test session (second session) of the FST were analyzed using the one-way ANOVA. When statistical differences were identified by ANOVA, the Bonferroni *post-hoc* test was used to assess specific differences between experimental groups. *p* < 0.05 was assumed as significant.

## 3. Results

### 3.1. Determination of the Microinjection Sites

Photomicrograph of a coronal brain section representing the bilateral microinjection sites in the MeA of a representative animal is presented in Figure 1. Diagrammatic representations based on Paxinos and Watson [40] depicting the microinjection sites in the MeA of all animals used in the present study are also shown in Figure 1.

### 3.2. Effects of MeA Treatment with Angiotensinergic Antagonists on Behavioral Changes in the Forced Swimming Test Evoked by Acute Restraint Stress

Immobility—analysis of immobility time at the pre-test session (first session, 15 min) indicated effect of time (F (1.804, 103.8) = 45.9, *p* < 0.0001) and treatment (F (4, 58) = 9.0, *p* < 0.0001), but without a time x treatment interaction (F (8, 115) = 0.9, *p* = 0.4635). Post-hoc analysis revealed that restraint stress decreased the immobility time in all period analyzed in animals treated with either vehicle (*p* < 0.0001), the selective AT_1_ receptor antagonist losartan (*p* < 0.0001) or the selective MAS receptor antagonist A-779 (*p* < 0.0001), when compared to the non-stressed group (Figure 2). The effect of restraint stress was not identified in animals subjected to microinjection of the selective AT_2_ receptor antagonist PD123319 into the MeA (*p* = 0.3567) (Figure 2). Besides, analysis indicated that immobility values of PD123319-treated animals were higher in relation to the vehicle group (*p* < 0.05) (Figure 2).

Regarding the immobility time during the test session (second session, 5 min), the analysis indicated differences between the experimental groups (F (4, 58) = 7.9, *p* < 0.0001) (Figure 2). Post-hoc analysis revealed that restraint stress decreased the immobility time in animals treated with either vehicle (*p* < 0.0001), losartan (*p* = 0.0034) or PD123319 (*p* = 0.0329) in the MeA, when compared to the non-stressed group (Figure 2). However, the effect of restraint stress was not identified in animals treated with A-779 into the MeA (*p* = 0.1179) (Figure 2).

Latency—analysis of latency to the first bout of immobility during the first session of the FST did not indicate differences between the experimental groups (F (4, 58) = 2.164, *p* = 0.0843) (Figure 2). Conversely, analysis during the second session indicated a significant effect (F (4, 58) = 2.828, *p* = 0.0375) (Figure 2). However, post-hoc analysis did not reveal specific difference between the experimental groups (*p* > 0.05) (Figure 2).

Swimming—analysis of swimming time during the first session did not indicate effect of either time (F (1.860, 103.8) = 1.510, *p* = 0.2263) or treatment (F (4, 58) = 2.46, *p* = 0.0557) or time x treatment interaction (F (8, 115) = 0.1770, *p* = 0.9936) (Figure 3). Conversely, analysis of swimming time during the second session indicated difference between the experimental groups (F (4, 58) = 4.074, *p* = 0.0075) (Figure 3). Post-hoc analysis revealed a decrease in the swimming time in animals treated with PD123319 (*p* = 0.0314) in relation to the vehicle group (Figure 3).

Climbing—analysis of climbing time during the first session indicated effect of time (F (1.860, 103.8) = 20.3, *p* < 0.0001), but without influence of treatment (F (4, 58) = 0.8928, *p* = 0.4742) and time x treatment interaction (F (8, 115) = 0.3281, *p* = 0.9537) (Figure 3). Analysis of climbing during the second session also indicated significant effect (F (4, 58) = 4.22, *p* = 0.0062) (Figure 3). However, post-hoc analysis did not reveal specific differences between the experimental groups (*p* > 0.05) (Figure 3).

## 4. Discussion

Results reported in the present study provide the first evidence of an involvement of MeA angiotensinergic neurotransmission in behavioral changes evoked by aversive threats. Indeed, we observed that exposure to a 60 min session of restraint session decreased the immobility during both the first (15 min session performed immediately after the restraint) and second (5 min session performed 24 h after the first session) sessions of the FST. However, we did not identify effect of restraint stress in latency to the first bout of immobility, as well as in swimming and climbing behaviors in the FST. Regarding the role of angiotensinergic receptors in the MeA, the treatment with the selective AT_2_ receptor antagonist PD123319 inhibited the restraint-evoked decrease in immobility during the first session, but without affecting the change during the second session. Furthermore, the effect of restraint in immobility during the second session was not identified in rats treated with the selective MAS receptor antagonist A-779 in the MeA. Finally, our results revealed a decrease in the swimming behavior in the second session of the FST in animals that received the selective AT_2_ antagonist into the MeA.

Previous studies provided controversial results regarding the effect of stress in behaviors in the FST. Indeed, either increase, decrease or absence of effect in immobility and others behavioral parameters in the FST were reported following exposure to an acute session of stress [4,5,6,54,55,56,57,58]. Some evidence indicated that these discrepant findings might be related to differences in the experimental protocols, such as interval between the end of stress session and the start of FST, housing conditions (collective x isolated), and intensity and characteristics (escapable or inescapable) of the aversive stimulus [4,54]. Age has also been reported as a potential factor affecting the stress effects on rodent behaviors in the FST [5].

Specifically regarding the restraint stress, it was reported decrease in immobility and increase in swimming behavior immediately after an acute 60 min session of restraint stress [4]. Nevertheless, opposite effect was observed after 2 h of acute restraint stress [6], thus suggesting that effects in the FST are affected by duration of restraint. Increase in immobility was also reported in the FST 40 min after a 7 h session of restraint [57,58], thus reinforcing the idea that longer exposition to restraint stress is related to increased immobility. Therefore, the findings obtained in the present study of decrease in immobility during both the first and second sessions of the FST corroborate results reported by Armario et al. [4]. However, to the best of our knowledge, our results are the first evaluating the behavior in the FST 24 h after an acute session of restraint in animals submitted to a previous session of forced swimming (i.e., pre-test session). Indeed, Armario et al. [4] did not identify effect of 60 min session of restraint in the FST 24 h later in animals that were not submitted to a previous session of forced swimming.

A role of MeA in behavioral responses in the FST is supported by evidence obtained in non-stressed animals that demonstrated decrease in immobility following local MeA treatment with imipramine or a selective V_1b_ receptor antagonist [32,34]. Regarding angiotensinergic receptors in the MeA, we reported recently that MeA treatment with losartan decreased the immobility time in the FST in non-stressed animals [42]. This study also revealed that either PD123319 or A-779 microinjection into the MeA did not affect the immobility, but altered the active behavioral pattern of climbing and swimming [42]. Thus, results reported in the present study provide new evidence regarding the control of behavioral responses in the FST by angiotensinergic receptors in the MeA. Findings documented here indicate an involvement of MeA angiotensinergic neurotransmission in restraint-evoked behavioral changes in the FST. Specifically, our results revealed that the decreased immobility evoked by restraint stress in the pre-test session is mediated, at least partly, by activation of AT_2_ receptors in the MeA. The absence of restraint stress effect in immobility in the second session in animals treated with A-779 in the MeA provides evidence for an involvement of the MAS receptor in this behavioral response.

Interestingly, comparison of the present results with those reported previously in non-stressed animals [42] indicates that MeA angiotensinergic neurotransmissions differently control behavior in the FST in naïve (non-stressed) and stressed animals. As mentioned above, we identified that MeA treatment with losartan 10 min before the pre-test session decreased the immobility time in the test session of the FST in non-stressed animals (behavior was not evaluated in the pre-test session) [42]. Conversely, the present study indicates an involvement of AT_2_ and MAS receptors in the MeA in the decreased immobility evoked by restraint stress, but without a role of local AT_1_ receptors. Although surprising, our results are supported by previous evidence of different effects of pharmacological treatments of limbic structures in the innate anxiety and anxiogenic response induced by restraint stress in the elevated plus maze (EPM) [59,60]. For instance, Campos et al. [59] reported that the anxiogenic effect observed in naïve animals in the EPM following ventral hippocampus treatment with an endocannabinoid reuptake inhibitor shifted to an anxiolytic effect in animals submitted to restraint stress (i.e., the endocannabinoid facilitation inhibited restraint-evoked anxiogenic response). Similar effect was observed with medial prefrontal cortex treatment with the phytocannabinoid cannabidiol [60]. We have also identified that bed nucleus of stria terminalis treatment with a CB_2_ cannabinoid receptor antagonist led to an anxiogenic effect in the EPM in naïve animals, whereas the same treatment inhibited the anxiogenic effect evoked by restraint stress [61]. Thus, as reported previously for behaviors related to anxiety, the results reported here taken together with those of Moreno-Santos et al. [42] indicate that behavioral control in the FST by angiotensinergic mechanisms in the MeA is influenced by previous stressful experience. In this sense, to the best of our knowledge, our findings are the first indicating an influence of stressful experience in neurobiological mechanisms regulating behavior in the FST.

The decreased swimming and tendency of increase in climbing (42% increase in relation to vehicle group) during the second session of the FST in animals treated with the selective AT_2_ receptor antagonist in the MeA are, at least partly, in line with previous results [42]. Indeed, this previous study identified increased climbing and a tendency of decreased swimming in non-stressed animals treated with PD123319 in the MeA. Previous pharmacological studies investigating the effects of antidepressant drugs in the FST provided evidence that active behaviors, such as swimming and climbing, seem to be regulated by specific monoaminergic encephalic circuits [50]. Indeed, these studies identified that selective serotonin reuptake inhibitors increased specifically the swimming behavior, whereas selective noradrenaline uptake inhibitors acted mainly increasing climbing [49,62,63,64,65]. Therefore, the control of active behaviors during the FST by AT_2_ receptor in the MeA might be mediated by stimulation of facilitatory afferences to serotoninergic circuits and/or inhibition of noradrenergic pathways. Facilitation of local serotoninergic neurotransmission and/or inhibition of noradrenergic mechanisms within the MeA might also contribute. Nevertheless, further studies are needed to clarify an interaction between MeA angiotensinergic neurotransmission and monoaminergic pathways in the control of behaviors in the FST.

In summary, the findings reported here indicate that an acute 60 min session of restraint stress decreases the immobility in the first (15 min) and second (5 min) sessions of the FST in rats. The decrease in immobility in the first session seems to be mediated by AT_2_ receptors activation in the MeA, whereas MAS receptors is potentially involved in the behavioral change evoked by restraint stress during the second session. Our results also confirm a role of MeA AT_2_ receptors controlling the pattern of active behaviors (i.e., climbing and swimming) during the FST.

## Figures and Tables

**Figure 1 cells-10-01217-f001:**
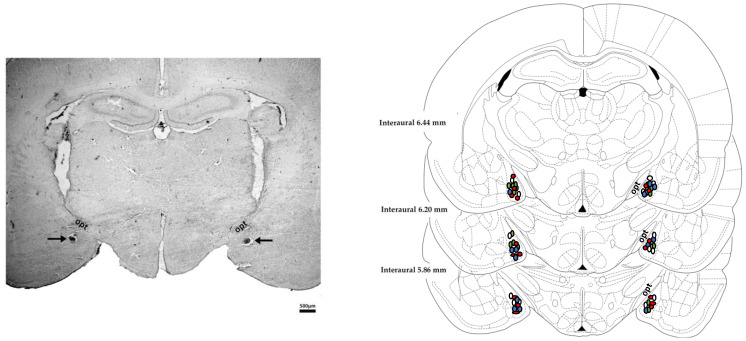
Microinjection sites into the MeA. (**Left**) Photomicrograph of a coronal brain section illustrating bilateral injection sites in the medial amygdaloid nucleus (MeA) of a representative animal. (**Right**) Diagrammatic representation based on Paxinos & Watson [40] showing the microinjection sites in the medial amygdaloid nucleus (MeA) of the selective AT_1_ receptor antagonist losartan (red circles), the selective MAS receptor antagonist A-779 (blue circles), the selective AT_2_ receptor antagonist PD123319 (green circles) and vehicle (white circles). The number shown in the figure can be smaller than the total number of animals used in this experiment due to the overlap of some injection sites. Opt—optical tract.

**Figure 2 cells-10-01217-f002:**
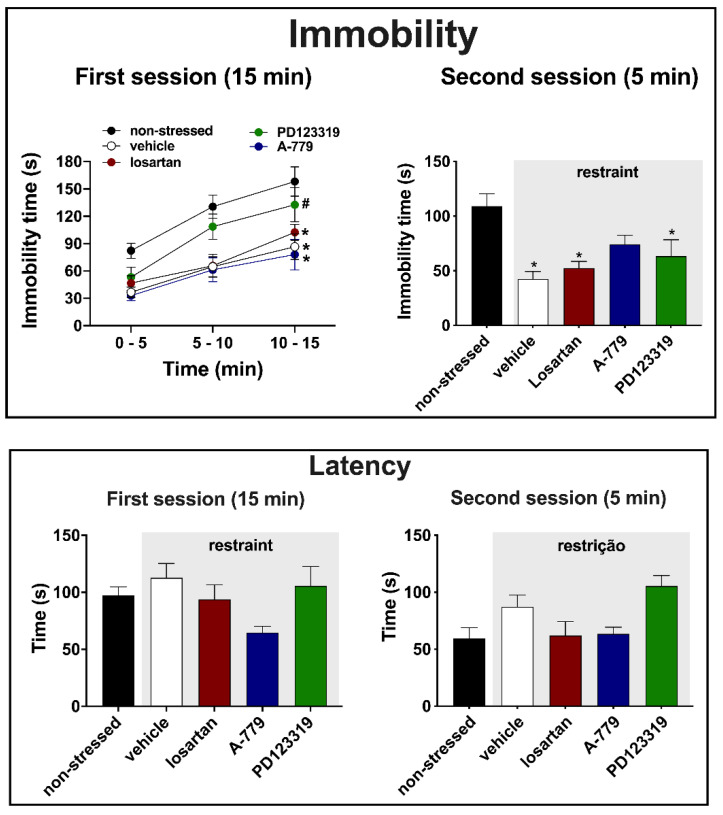
Immobility time and latency to the first bout of immobility in the forced swimming test (FST) of animals submitted to an acute session of restraint stress and that received bilateral microinjection into the medial amygdaloid nucleus (MeA) of the selective AT_1_ receptor antagonist losartan (1 nmol/100 nL, *n* = 9), the selective MAS receptor antagonist A-779 (0.1 nmol/100 nL, *n* = 10), the selective AT_2_ receptor antagonist PD123319 (5 nmol/100 nL, *n* = 8) or the vehicle (saline, 100 nL, *n* = 21); as well as animals who were not submitted to restraint stress (non-stressed, *n* = 15). (**Top**) Immobility time each 5 min during the pre-test session (first session, **left**), which was performed immediately after the end of the acute restraint session, and during the entire test session (second session, **right**) that occurred 24 h after the pre-test session of the FST. The bars and circles represent the mean ± SEM. Results of the first session were analyzed using two-way ANOVA followed by Bonferroni post-hoc test, whereas one-way ANOVA was used in the results of the second session. * *p* < 0.05 vs. non-stressed group (during the entire period for the results of the first session); # *p* < 0.05 vs. vehicle group (subjected to restraint stress) during the entire period. (**Bottom**) Latency to the first bout of immobility during the first and second sessions of the FST. The bars represent the mean ± SEM. One-way ANOVA followed by Bonferroni post-hoc test.

**Figure 3 cells-10-01217-f003:**
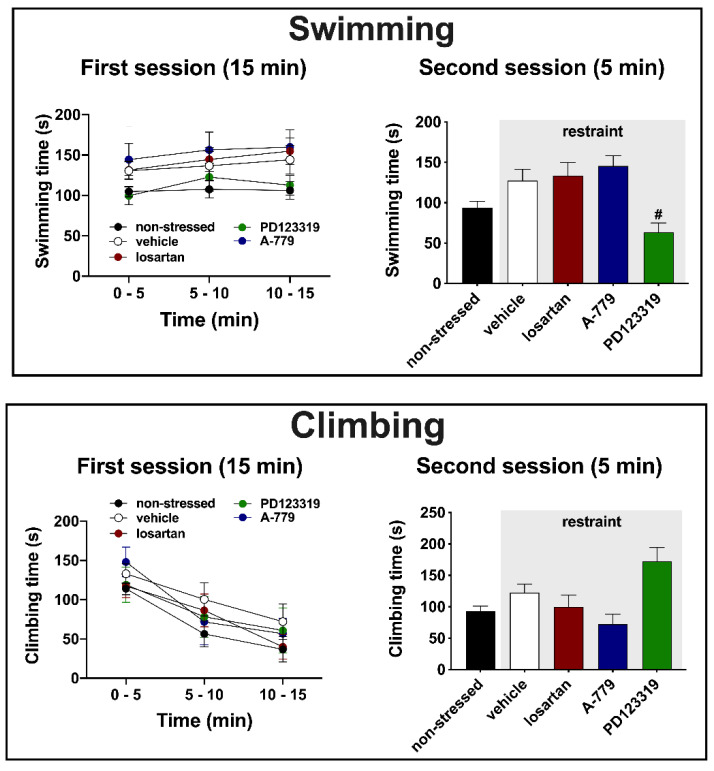
Swimming and climbing time in the forced swimming test (FST) of animals submitted to an acute session of restraint stress and that received bilateral microinjection in the medial amygdaloid nucleus (MeA) of the selective AT_1_ receptor antagonist losartan (1 nmol/100 nL, *n* = 9), the selective MAS receptor antagonist A-779 (0.1 nmol/100 nL, *n* = 10), the selective AT_2_ receptor antagonist PD123319 (5 nmol/100 nL, *n* = 8) or the vehicle (saline, 100 nL, *n* = 21); as well as animals who were not submitted to restraint stress (non-stressed, *n* = 15). (**Top**) Swimming time each 5 min during the pre-test session (first session, **left**), which was performed immediately after the end of the acute restraint session, and during the entire test session (second session, **right**) that occurred 24 h after the pre-test session of the FST. (**Bottom**) Climbing time each 5 min during the pre-test session (first session, **left**), and during the entire test session (second session, **right**). The bars represent the mean ± SEM. Results of the first session were analyzed using two-way ANOVA, whereas one-way ANOVA followed by Bonferroni post-hoc test was used in the results of the second session. # *p* < 0.05 vs. vehicle group (subjected to restraint stress).

## Data Availability

All relevant data are presented.

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
