# Peer review of "Angiotensinergic Neurotransmissions in the Medial Amygdala Nucleus Modulate Behavioral Changes in the Forced Swimming Test Evoked by Acute Restraint Stress in Rats"

_cells, 2021, doi:10.3390/cells10051217_

Round 1
Reviewer 1 Report
In this study, the role of the AT receptors in the medial amygdala in stress-induced behavioral changes in the forced swimm test is investigated. The authors show that a blockade of the Mas receptor reduced acute changes after stress while AT2 receptor blockade reduced the changes 24 h later. The article is well written but would benefit from some language editing. The effects are robust but could be analysed and illustrated a bit different. Last, it is difficult to understand what the stress-induced behavioral changes shown here mean. Are these adaptive or depression-like changes?
Please see my specific comments:
(1) Please introduce all abbreviations in the abstract (AT, Mas, MeA,...).
(2) In general, the article would benefit from a careful language editing.
(3) Consider to rephrase the sentence starting in line 36.
(4) What is meant by "generation" in line 42? Synthesis?
(5) Line 58: Please mention the MoA of minaprine.
(6) Losartan was used in a potassium salt formulation. Is it possible thar the potassium ions induce some effects? Usually, neural activity is very sensitive to changes in extracellular potassium concentrations.
(7) Please explain how behavior was measured or scored. If manually, make a statement whether observer were blinded.
(8) Please insert spaces before and after mathematical signs (=, <, ...).
(9) What do degrees of freedoms with decimals mean? (line 186).
(10) Line 185ff: IN my opinion, there is no need to perform all these post-hoc tests if there is no time x treatment interaction. Why not performing a post-hoc test only for the treatment effect ("main column effects" in Prism if columns are treatment)?
(11) Why not showing the data with line diagrams, as it is usual for time courses. This would also reduce the figure sizes a bit and would fit to the analysis suggested in the previous comment.
(12) The Brazilian heading in Figure 2/Latency should be replaced by a English one ("restraint").
(13) Line 271ff: How do the authors interpret a decrease of immobility after stress? Is this a depression-like effect? Or an adaptive response?
(14) Line 330ff: Consider rephrasing this sentence. It's difficult to understand.
(15) Line 342ff: "indicate". I don't think that the data indicate this. This is "just" a potential underlying mechanisms.
Author Response
Reviewer 1 comments
We thank you for the helpful observations. All suggestions were accepted, and corresponding changes were included in the new version of the manuscript. A point-by-point reply to the comments is presented below.
Yours sincerely,
Carlos C. Crestani, PhD
(1) Please introduce all abbreviations in the abstract (AT, Mas, MeA,...).
RESPONSE: All abbreviations presented in the abstract were defined in the revised version of the manuscript. Abbreviations of angiotensin receptors in the abstract and throughout the manuscript were used as stablished by the International Union of Basic and Clinical Pharmacology (Gasparo et al, Pharmacol Rev 52(3):415-72, 2000; Karnik et al, Pharmacol Rev 67(4):754-819, 2015).
(2) In general, the article would benefit from a careful language editing.
RESPONSE: The manuscript was carefully revised, and mistakes were corrected throughout the text.
(3) Consider to rephrase the sentence starting in line 36.
RESPONSE: As suggested, the sentence was revised in the new version of the manuscript (please, see lines 64-66 of the revised manuscript).
(4) What is meant by "generation" in line 42? Synthesis?
RESPONSE: In fact, we mean “synthesis”. The mistake was corrected in the new version of the manuscript (please, see line 69 of the revised manuscript).
(5) Line 58: Please mention the MoA of minaprine.
RESPONSE: Information was included as suggested (please, see line 85 of the revised manuscript).
(6) Losartan was used in a potassium salt formulation. Is it possible thar the potassium ions induce some effects? Usually, neural activity is very sensitive to changes in extracellular potassium concentrations.
RESPONSE: The issue is not easy to address once the vast majority of studies does not mention the losartan formulation salt. However, a previous electrophysiological study demonstrated that losartan potassium did not affect basal K+ current in CATH.a cells (Sun et al, Biochem Biophys Res Commun 310(3):710-4, 2003). However, this same study identified that losartan potassium inhibited the changes in K+ current evoked by angiotensin II, thus indicating that effects of this antagonist are mediated by blockade of AT1 receptors rather than electrophysiological changes.
The idea that losartan potassium does not affect neural activity within the MeA is further supported by results of the present study indicating that microinjection of this drug within the MeA did not affect restraint-evoked behavioral changes in the FST. Besides, we have previously reported that losartan microinjection into the MeA did not affect basal parameters of blood pressure and heart rate (Costa-Ferreira et al, Pflugers Arch. 471(9):1173-1182, 2019; and Eur J Neurosci. 53(3):763-777, 2021).
(7) Please explain how behavior was measured or scored. If manually, make a statement whether observer were blinded.
RESPONSE: The behaviors in the FST were manually analyzed in a blind manner. The software X-Plot-Rat was used to help calculate the score of each behavior. These information are presented in the revised version of the manuscript (please, see section 2.5. Forced swimming test)
(8) Please insert spaces before and after mathematical signs (=, <, ...).
RESPONSE: Changes were included as suggested in the revised version of the manuscript.
(9) What do degrees of freedoms with decimals mean? (line 186).
RESPONSE: Due to the limited number of points in the time course curve, the method of Greenhouse and Geisser was used to adjust the results of the repeated measures ANOVA to account for the value of epsilon. The only thing this adjustment does is reduce the number of degrees of freedom (values can be fractional) for the factor time, which increases the P value.
(10) Line 185ff: In my opinion, there is no need to perform all these post-hoc tests if there is no time x treatment interaction. Why not performing a post-hoc test only for the treatment effect ("main column effects" in Prism if columns are treatment)?
RESPONSE: Changes were included as suggested in the revised version of the manuscript (please, see lines 382-384 of the revised manuscript).
(11) Why not showing the data with line diagrams, as it is usual for time courses. This would also reduce the figure sizes a bit and would fit to the analysis suggested in the previous comment.
RESPONSE: As suggested, data of first session of the FST are shown with line diagrams in the new Figures 1 and 2.
(12) The Brazilian heading in Figure 2/Latency should be replaced by a English one ("restraint").
RESPONSE: The mistake was corrected in the revised version of the manuscript (please, see the new Figure 2).
(13) Line 271ff: How do the authors interpret a decrease of immobility after stress? Is this a depression-like effect? Or an adaptive response?
RESPONSE: The question is very pertinent. In fact, previous studies evaluating the effect of stressful stimuli in behavior in the FST have related these changes to depression-like state (e.g., Armario et al, Pharmacol Biochem Behav 39(2):373-7, 1991; Platt & Stone, Eur J Pharmacol 82(3-4):179-81, 1982; Sevgi et al, Methods Find Exp Clin Pharmacol 28(2):95-9, 2006; Moretti et al, J Mol Neurosci 49(1):68-79, 2013; Bettio et al, Pharmacol Biochem Behav 127:7-14, 2014; Almeida et al, Stress. 18(4):462-74, 2015; McNeal et al, Stress 20(2):175-182, 2017; Cotella et al, Prog Neuropsychopharmacol Biol Psychiatry. 88:303-310, 2019). Therefore, the decrease in immobility in the FST identified in the present study might be interpreted as an adaptive response (Platt & Stone, Eur J Pharmacol 82(3-4):179-81, 1982).
(14) Line 330ff: Consider rephrasing this sentence. It's difficult to understand.
RESPONSE: The sentence was revised in the new version of the manuscript (please, see lines 630-632 of the revised manuscript).
(15) Line 342ff: "indicate". I don't think that the data indicate this. This is "just" a potential underlying mechanisms.
RESPONSE: As suggested, the text was revised in order to discuss that modulation of monoaminergic mechanisms are potential mechanisms involved in regulation of active behaviors in the FST by AT2 receptors within the MeA (please, see lines 643-647 of the revised manuscript).

Reviewer 2 Report
This manuscript investigated the role of the renin-angiotensin system in acute stress response in rats. Specific AT1, AT2, and Mas receptor antagonists were administered i.c.v. in the medial amygdala, followed by 60 min of restraint stress. Behavioral responses were then investigated immediately and 24 hours after in the forced swim test. The setups of the experiments are unconventional, and the rationales for their use are not explained well. The technical execution of the i.c.v. injections and behavioral studies are well-described. Therefore, I do not have concerns about the experiments themselves, but I have major concerns about the selection of the models and how the introduction and discussion have been written given the models.
Major comments:
- This section concentrates largely on depression and depression-like behavior. It is problematic because the model that was used in the study, the acute restraint stress, is not a validated model for depression. Therefore, the introduction should concentrate on the renin-angiotensin system in acute stress. Also, please introduce the system better in the second paragraph, i.e. what are the component receptors of the system, where they are expressed, what are their functions etc. This would help a reader who is not familiar with the system.
- The forced swim test. The authors have used a very unconventional version of the test. More rationale is needed to understand why a pre-test for 15 min was carried out immediately after the restraint stress and a 5 min test 24 hours later. How are these protocols validated?
- This section is somewhat difficult to read and should be simplified. What is the rationale of comparing drug-treated animals with non-stressed controls? They should only be compared with vehicle-treated animals because cannulation and injection constitute additional stressors.
- Figure 2. Statistical comparisons and associated p-values of the drug groups should be shown in comparison to the vehicle group only.
- Because the authors used an unconventional 15-min FST, they should indicate in the discussion when the discussed papers have used this particular model and when a more conventional 6-min duration was used. The duration may have a large effect on the results and consequently on the interpretation of the results.
Author Response
Reviewer 2 comments
We thank you for the helpful observations. All suggestions were accepted, and corresponding changes were included in the new version of the manuscript. A point-by-point reply to the comments is presented below.
Yours sincerely,
Carlos C. Crestani, PhD
Major comments:
This section concentrates largely on depression and depression-like behavior. It is problematic because the model that was used in the study, the acute restraint stress, is not a validated model for depression. Therefore, the introduction should concentrate on the renin-angiotensin system in acute stress. Also, please introduce the system better in the second paragraph, i.e. what are the component receptors of the system, where they are expressed, what are their functions etc. This would help a reader who is not familiar with the system.
RESPONSE: We appreciate the comment. In fact, restraint stress is not a specific model for depression. However, it was used in the present study as a stressor stimulus that evokes behavioral changes in the FST related to depression-like state (please, see reply to issue # 13 of the Reviewer 1). As suggested, a paragraph describing the renin-angiotensin system (RAS) was included in the Introduction section of the new version of the manuscript.
The forced swim test. The authors have used a very unconventional version of the test. More rationale is needed to understand why a pre-test for 15 min was carried out immediately after the restraint stress and a 5 min test 24 hours later. How are these protocols validated?
This section is somewhat difficult to read and should be simplified. What is the rationale of comparing drug-treated animals with non-stressed controls? They should only be compared with vehicle-treated animals because cannulation and injection constitute additional stressors.
RESPONSE: The procedure of FST used in the present study was as described by Porsolt et al. for rats (Nature 266 (5604):730-2, 1977; Eur J Pharmacol 47(4):379-91, 1978). Unlike mice, in which the test was stablished with only one 6 min session (Porsolt et al, Arch Int Pharmacodyn Ther 229(2):327-36, 1977), the FST in rats was described in two sessions (15 min session followed by 5min session 24h later). However, in fact, several previous studies in rats evaluating the impact of stressful stimuli in the FST have investigated the behavior in only one 5 min session of forced swimming performed either immediately or 24 h after ending the stress session (Armario et al, Pharmacol Biochem Behav 39(2):373-7, 1991; Platt & Stone, Eur J Pharmacol 82(3-4):179-81, 1982; Sevgi et al, Methods Find Exp Clin Pharmacol 28(2):95-9, 2006; Bernal-Morales et al, Behav Processes ;82(2):219-22, 2009). Nevertheless, it is worth mentioning that a previous study used protocol similar in relation to that of the present study, in which rats were subjected to a 15 min session of forced swimming after restraint stress (2.5h duration) followed by a 5 min session 24h later (Platt & Stone, Eur J Pharmacol 82(3-4):179-81, 1982). Besides, we have previously used a protocol with two sessions of forced swimming for evaluation of the effect of a chronic variable stress protocol in behavior in the FST (Almeida et al, Stress. 18(4):462-74, 2015).
Regarding the non-stressed animals, this group was included to allow us to determinate the impact of restraint stress experience in behavior in the FST. In this sense, it is worth mentioning that we have identified during standardization of the model that the restraint-evoked behavioral changes in the FST in animals subjected to intra-brain microinjection of saline 10 min before restraint onset was similar in relation to naïve animals (i.e., animals submitted to restraint that was not subjected to stereotaxic surgery and intra-brain microinjection), thus indicating that intra-brain canula implantation and injection do not affect the behavioral changes in the FST evoked by restraint stress.
The text was revised to clarify the rationale of the FST in two sessions and the inclusion of a non-stressed group.
Figure 2. Statistical comparisons and associated p-values of the drug groups should be shown in comparison to the vehicle group only.
Because the authors used an unconventional 15-min FST, they should indicate in the discussion when the discussed papers have used this particular model and when a more conventional 6-min duration was used. The duration may have a large effect on the results and consequently on the interpretation of the results.
RESPONSE: Regarding the statistical comparisons in the Figure 2, the comparison of groups subjected to restraint stress with the non-stressed group is important to determinate the effect of restraint stress in behavior in the FST. The same approach has been used in previous studies that evaluated the anxiogenic effect of restraint stress in the elevated plus maze (e.g., Scopinho et al, PLoS One 8(10):e77750, 2013; Gouveia et al, Neuropharmacology 101:379-88, 2016; Busnardo et al, Prog Neuropsychopharmacol Biol Psychiatry. 90:16-27, 2019).
In fact, differences in experimental protocol for evaluation of stress-evoked behavioral changes in the FST have a prominent influence on the results. In this sense, the results documented in the present study are discussed in detail in relation to previous studies in the paragraphs 2 and 3 of the Discussion section.

Round 2
Reviewer 2 Report
Thank you for clarifications and adding them to the revised version of the manuscript. I now understand the rationale of the work, and do not have any further suggestions. The manuscript has improved considerably upon revision.